# Performance Assessment of Concrete Using Discarded Membrane Filter Materials

**Sehwan Park and Junkyeong Kim ***

Safety Inspection for Infrastructure Laboratory (SIIL), Advanced Institute of Convergence Technology (AICT), 145 Gwanggyo-ro, Yeongtong-gu, Suwon-si 16229, Korea; sehwan0721@snu.ac.kr
* Correspondence: junkyeong@snu.ac.kr; Tel.: +82-31-888-9524

**Abstract:** Currently, membrane filters, which need to be replaced regularly as they get worn, are used in filtration facilities globally. The old membrane filters and housings become continuous industrial waste and are currently 100% incinerated. To solve this environmental problem, this study proposes the development of an eco-friendly concrete by mixing waste membrane resources with concrete. Through this, the environmental pollution and wastage of resources due to incineration, and the enormous amount of carbon dioxide generated during cement production, can be decreased by reducing the cement required when mixing concrete. To this end, the membrane module outer surface (acrylic butadiene styrene, ABS) and inner membrane (poly vinylidene fluoride, PVDF) were extracted from the waste membrane system and pulverized. Different mix ratios of 1%, 3%, and 5% for replacing cement were used when mixing concrete. The test specimens were then tested and compared with the reference concrete (ordinary Portland cement) specimen. It was confirmed that the compressive strength was high after 28 days in all the specimens to which ABS was added at 1%, 3%, and 5% mix ratios. Therefore, the possibility of technological development of eco-friendly concrete using waste resources from membrane filtration facilities was verified.

**Keywords:** membrane filter recycling; ordinary Portland cement (OPC); poly acrylic butadiene styrene (ABS); poly vinylidene fluoride (PVDF); eco-friendly concrete

## 1. Introduction

Currently, membrane filtration facilities and systems are continuously operated, with old membranes and housings being replaced when required. The old membranes and housings, which are being constantly generated, become industrial wastes that are 100% incinerated. These materials are made of plastic materials that, when incinerated, lead to environmental problems such as poor use of resources and pollution of the atmospheric environment. The issue of insufficient incineration and recycling technology of the old membrane materials (such as the separator and housing materials) is one that should be addressed. In other words, it is necessary to develop recycling technologies for each material involved in the process in order to prevent environmental pollution caused by the incineration of membrane waste.

Reusing these waste resources in concrete has the effect of reducing the cost of treating waste resources, protecting the environment from pollution, and has many advantages, such as improving some properties of concrete [1]. Therefore, research on eco-friendly concrete technology using waste resources is being actively conducted to prevent environmental pollution and reduce carbon emissions. Strength measurements and performance evaluation were performed by adding glass to concrete [2]; studies on a method using waste concrete were also conducted [3–7]. In particular, studies were actively conducted to investigate properties when using waste resources of plastic materials such as aged membranes and housings for concrete. By classifying plastics by type, strength and

performance evaluations were conducted by increasing the addition ratio when mixing concrete [8–11]; when plastics were used in concrete, the compressive strength and bonding strength decreased as the plastic was added, regardless of the type of plastic [12,13], but it was confirmed that the impact resistance and ductility increased [14]. In addition, it was confirmed that round plastic particles improved the workability of concrete, whereas angled plastic particles decreased the workability of concrete [15].

However, in the existing studies, the amount of cement was generally the same, and the experiment was carried out by increasing the ratio by replacing the aggregate used when mixing concrete [16]. When 1 ton of cement is generated in the production process, about 0.7–1.0 tonnes of carbon dioxide is emitted, accounting for 7–8% of global greenhouse gas emissions [17].

In Korea, the cement industry is accountable for approximately 56.7 million tons of carbon dioxide per year [18]. To solve this emission problem, research is being conducted to reduce the amount of cement in concrete by adding admixtures such as fly ash and blast furnace slag powder [19–21].

This study investigates eco-friendly concrete technology that utilizes waste membrane resources to reduce the amount of cement in concrete. In particular, workability and strength, which are the biggest problems of concrete with plastic added, were selected as the scope of the study. To achieve this, a comparison was made between ordinary concrete, ordinary Portland cement (OPC), and concrete that replaced an amount of cement with the waste membrane material. To ease the comparison, the ratio of fine aggregates, coarse aggregates, admixtures, and w/c (water–cement ratio) were kept constant. In order to determine the strength characteristics of the generated specimens, strength measurements were performed using universal test machine (UTM) equipment, and performance evaluation was conducted by comparing the strength of ordinary concrete with concrete using different ratios of waste membrane resources.

## 2. Material Preparation for Membrane Recycled Concrete

### 2.1. Material Preparation

The cement used in the experiment was Portland cement of KS L 5201 standard, with a specific gravity of 3.15 $t/m^3$ and a fineness of 3450 $cm^2/g$. Crushed aggregates with a maximum particle diameter of 25 mm were used as the coarse aggregate, and river sand was used as the fine aggregate.

The used membrane film and housing were each composed of plastic made of poly vinylidene fluoride (PVDF) and poly acrylic butadiene styrene (ABS). As an excellent chemical-resistant material, PVDF has a wide range of applications; moreover, it has the advantage of being physiologically safe. ABS has a high heat distortion temperature and excellent chemical and oil resistance.

The outer surface of the membrane module (ABS) and the inner membrane (PVDF) were extracted from the membrane filtration facility. To use the ABS and PVDF as a substitute of cement, they should be made into fine-grained powder. As such, the extracted ABS and PVDF were cut to a certain size and then frozen with liquid nitrogen. They were then pulverized to a thickness of 0.5 mm using pulverizing equipment as shown in Figure 1.

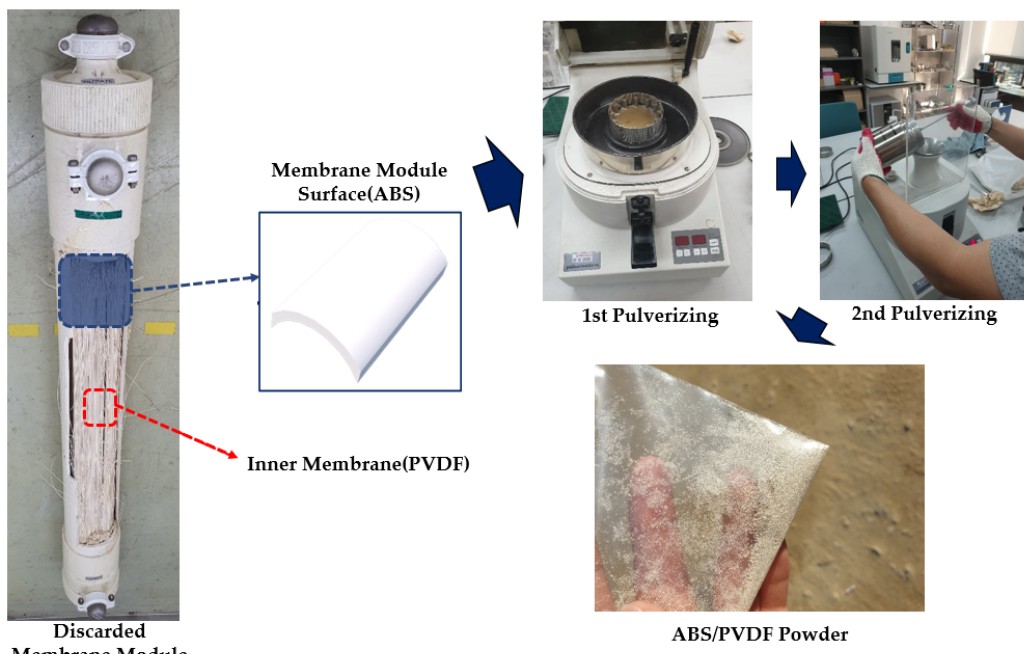

**Figure 1.** Extraction of ABS(poly acrylic butadiene styrene) and PVDF(poly vinylidene fluoride) powder from discarded membrane.

### 2.2. Mortar Test of ABS/PVDF Mixture

Before proceeding with the concrete mixing test of the sample material, a mortar test specimen was prepared by substituting the prepared powder with a certain ratio of cement to obtain a replaceable percentage. Tables 1 and 2 present the physical and chemical properties of the prepared ABS and PVDF powder.

**Table 1.** Properties of ABS materials.

| Relative Density | Water Absorption Rate (%) | Tensile Strength (MPa) | Bending Strength (MPa) | Rockwell Hardness | Heat Distortion Temperature (°C) |
|---|---|---|---|---|---|
| 1.02–1.05 | 0.2–0.45 | 35–44 | 51–81 | R65–109 | 93–103 |

**Table 2.** Properties of PVDF materials.

| Relative Density | Water Absorption Rate (%) | Tensile Strength (MPa) | Bending Strength (MPa) | Rockwell Hardness | Heat Distortion Temperature (°C) |
|---|---|---|---|---|---|
| 1.78 | 0.02 | 52–54 | 70–75 | R84 | 166–170 |

The mix proportions of the mortar specimens are shown in Table 3. ABS and PVDF powders were added to the mortar specimens for testing. Powders were replaced with 0%, 8%, 15%, and 25% cement. A 50 mm cube-shaped mortar test specimen was manufactured, and a compression test was conducted.

**Table 3.** Mortar specimen mix proportions.

| C:S (Weight Ratio) | W/C (%) | Unit Amount (kgf/m³) | | |
|---|---|---|---|---|
| | | Water (W) | Cement (C) | Fine Aggregate (S) |
| 1:2.5 | 50 | 255 | 566 | 1430 |

The results are shown in Table 4 and Figure 2. As the content of powder increased, the compressive modulus of elasticity increased by 20% compared to ordinary cement,

and the compressive strength gradually decreased. However, the possibility of replacing cement by approximately 8% was confirmed.

**Table 4.** Powder-containing mortar compression test results.

| Waste Membrane Powder (%) | Modulus (GPa) | Strength (kgf/cm²) | Peak Strain |
|---|---|---|---|
| 0 | 6.15 ± 0.15 | 557 ± 38.9 | 9.74 ± 1.31 |
| 8 | 7.84 ± 0.17 | 464 ± 40.6 | 8.26 ± 0.65 |
| 15 | 7.43 ± 0.18 | 314 ± 54.7 | 6.33 ± 0.55 |
| 25 | 5.39 ± 0.33 | 107 ± 22.9 | 5.00 ± 0.20 |

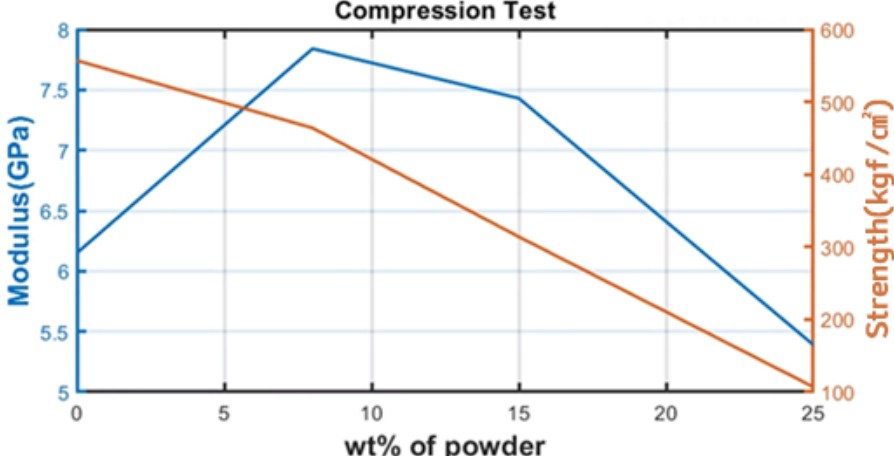

**Figure 2.** Powder-containing mortar compression test result graph.

## 3. Experimental Study

### 3.1. Preparation of OPC, ABS, and PVDF Concrete

The specimens were mixed on site based on the sieve analysis results shown in Table 5, according to the regulations of KS F 2502. In addition, fine aggregate, coarse aggregate, and compounded water were corrected according to (*a–d*) in Table 5 and Equations (1)–(5).

$$S' = \frac{S - b \times (S + G)}{1 - (a + b)} \tag{1}$$

$$G' = \frac{G - a \times (S + G)}{1 - (a + b)} \tag{2}$$

In Equation (1), $S'$ is the correction considering the grain size of fine aggregate, $S$ is fine aggregate, and $G$ is coarse aggregate. In (2), $G'$ is the correction considering the grain size of the coarse aggregate.

$$S'' = (1 + c) \times S \tag{3}$$

$$G'' = (1 + d) \times G' \tag{4}$$

$$W'' = W - (c \times S' + d \times G') \tag{5}$$

In (3), $S''$ is the correction considering the surface moisture of fine aggregate, and $G''$ in (4) is the correction considering the surface moisture of coarse aggregate. $W''$ in (5) is the correction considering the surface moisture of the blended water, and W is the blended water.

**Table 5.** Aggregate sieving test results.

|  | Specific Gravity | No.4 Sieve Remaining Amount | No.4 Sieve Passing Amount | Surface Water–Content Ratio |
|---|---|---|---|---|
| Fine aggregate | 2.58 | 1.0% | 99.0% | 2.7% |
| Coarse aggregate | 2.65 | 99.0% | 1.0% | −0.3% |

Based on the calibrated values, the job mix was performed, and cylindrical concrete specimens were manufactured according to the KS F 2405 standard (concrete compressive strength test method). Table 6 show the specified and job mix for each specimen; the unit of the values in the tables represents the unit amount per cube.

**Table 6.** Addition of ABS, PVDF powder (0%, 1%, 3%, 5%) in specified and job mix specimen (W: blended water, C: cement, S: fine aggregate, P: powder, G: coarse aggregate, AD: admixture).

|  | Unit (kg/m²) | | | | | |
|---|---|---|---|---|---|---|
|  | **W** | **C** | **S** | **P** | **G** | **AD** |
| Specified mix | 167 | 383(0%)<br>379(1%)<br>372(3%)<br>364(5%) | 789 | 4(1%)<br>11(3%)<br>19(5%) | 955 | 2.68 |
| Job mix | 149 | 383(0%)<br>379(1%)<br>372(3%)<br>364(5%) | 808 | 4(1%)<br>11(3%)<br>19(5%) | 954 | 2.68 |

In specimens to which the waste membrane resource was added, cement was replaced in increments of 1%, 3%, and 5%, within 8% of the standard set in the previous experiment.

To find the applicability of ABS and PVDF concrete, the air contents test, slump test and compressive strength test were performed, as shown in Figure 3. The air contents test and slump test were performed on the day of mixing concrete, and the compressive test was performed on the 3rd, 7th, and 28th day after mixing. All the tests were performed with ABS, PVDF concrete and OPC concrete to compare with ordinary concrete.

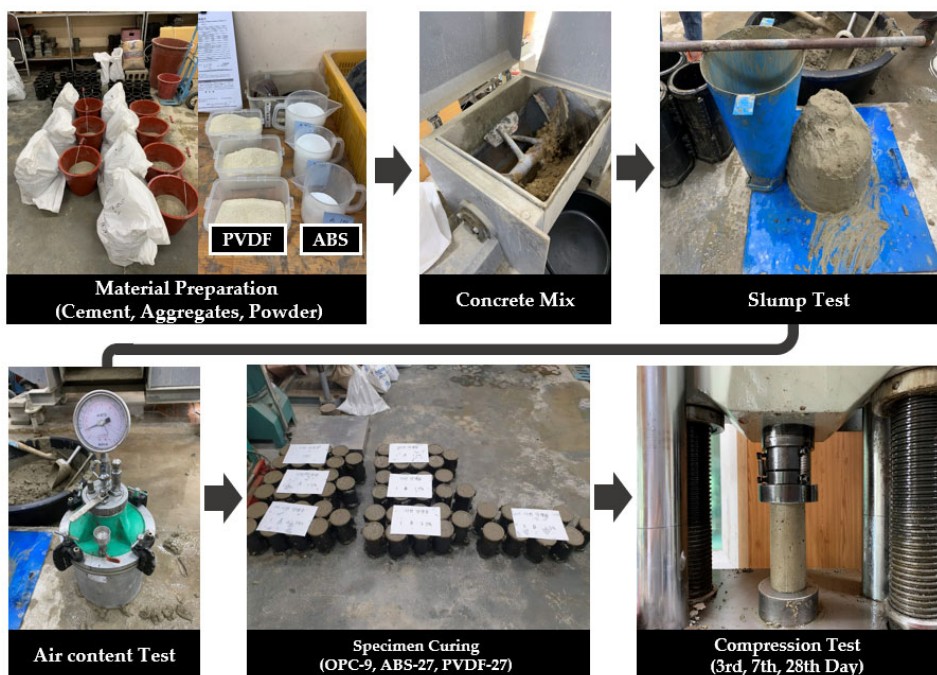

**Figure 3.** Testing process for ABS/PVDF concrete.

*3.2. Experimental Results*

The mix for the concrete was added in the order of fine aggregate, cement, coarse aggregate, powdered ABS, and PVDF powder. After mixing, the slump and air content were checked according to the specifications of the KS F 2402 (slump test method of Portland cement concrete) and KS F 2421 (air content of hardened concrete by pressure test method) standards.

As shown in Figure 4, the air content for the OPC specimen, ABS powder specimen, and PVDF powder specimen was equal to 9%. The slump content of the OPC and ABS powder specimens was the same, at 170 mm, but the slump content of the PVDF powder specimen was measured to be approximately 100 mm. The suitable slump of concrete which is used for reinforced specimen is 60–180 mm [22]. The result of the slump test of the ABS powder specimen was suitable for standard use, but the result of the PVDF powder specimen was higher than standard slump. This was caused by the high chemical resistance of PVDF, which was derived by the fluoride.

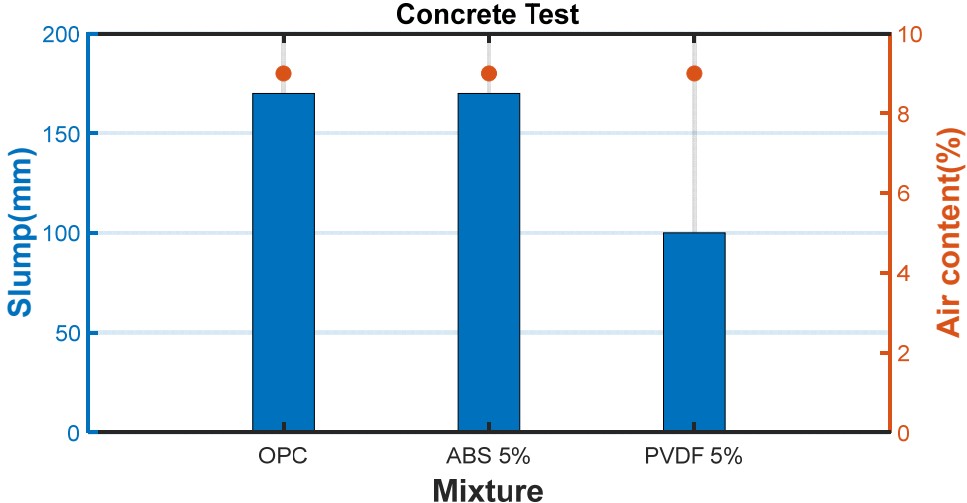

**Figure 4.** Slump and air content (Dot: air content; bar: slump).

The concrete specimens were cured in water, and strength measurements were performed according to the KS F 2405 standard using UTM equipment to measure the compressive strength of the prepared concrete by age.

Strength measurements were performed for each specimen, every 3 days, 7 days, and 28 days. The strength was measured for three specimens for each age, and the average value was derived. By comparing the average value of the compressive strength of the samples with ABS powder, PVDF powder, and the average value of the compressive strength of the OPC sample, the usability of the waste membrane was examined.

Figures 5–11 present graphs showing the strength measurement experiments for the three specimens of OPC, ABS, and PVDF. Additionally, the average strength values of the three specimens are shown together in the graphs, so OPC, ABS, and PVDF specimens can be compared using the average strength values.

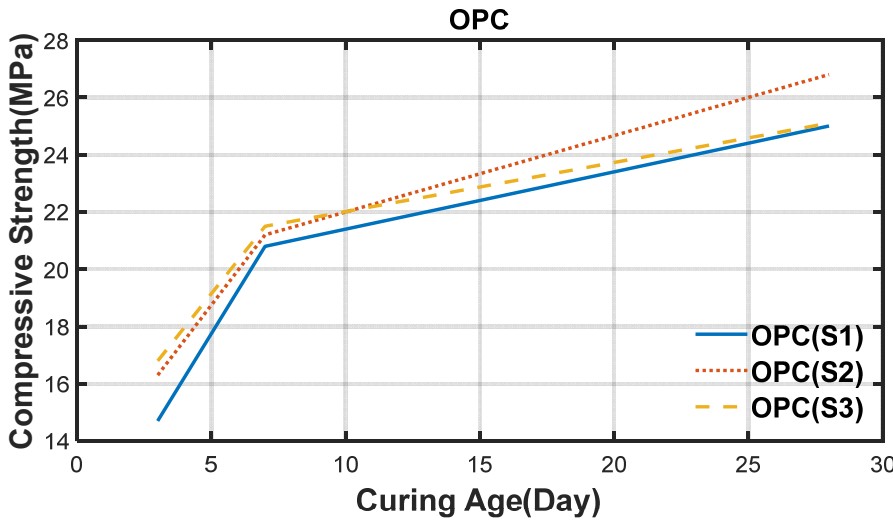

**Figure 5.** Graph of strength measurement results between OPC specimens.

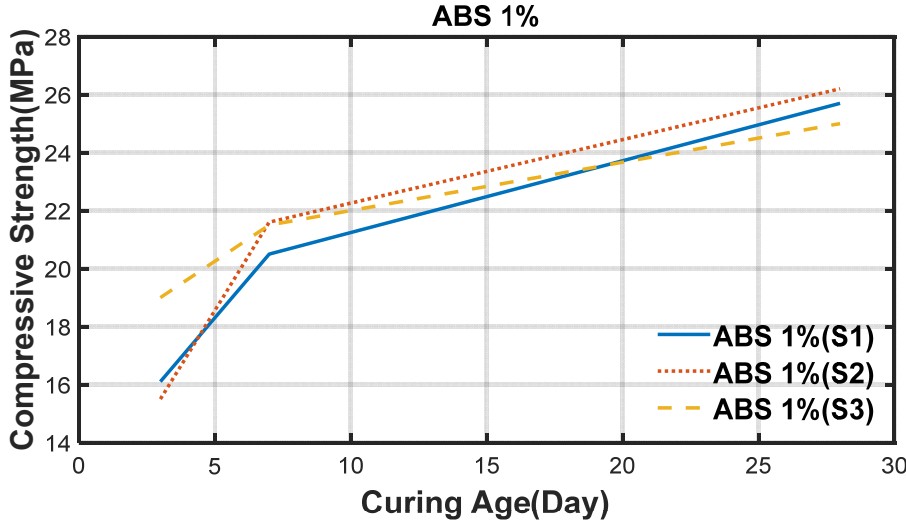

**Figure 6.** Graph of strength measurement results between ABS (1%) specimens.

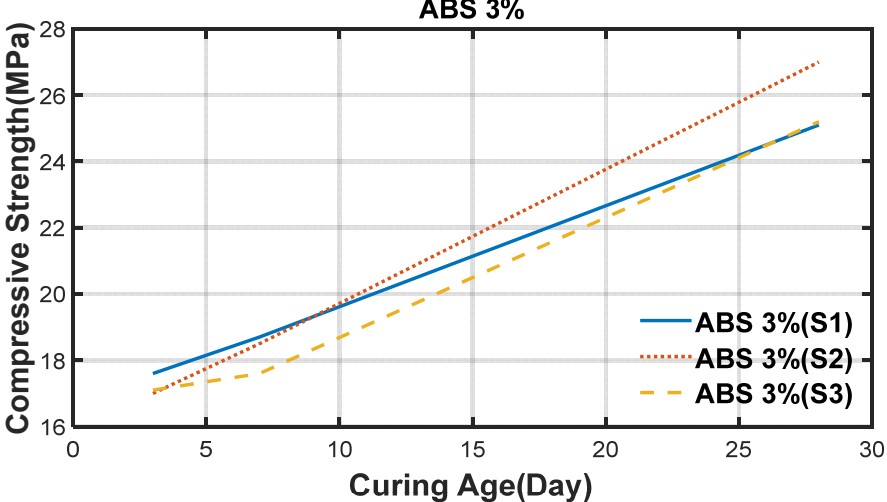

**Figure 7.** Graph of strength measurement results between ABS (3%) specimens.

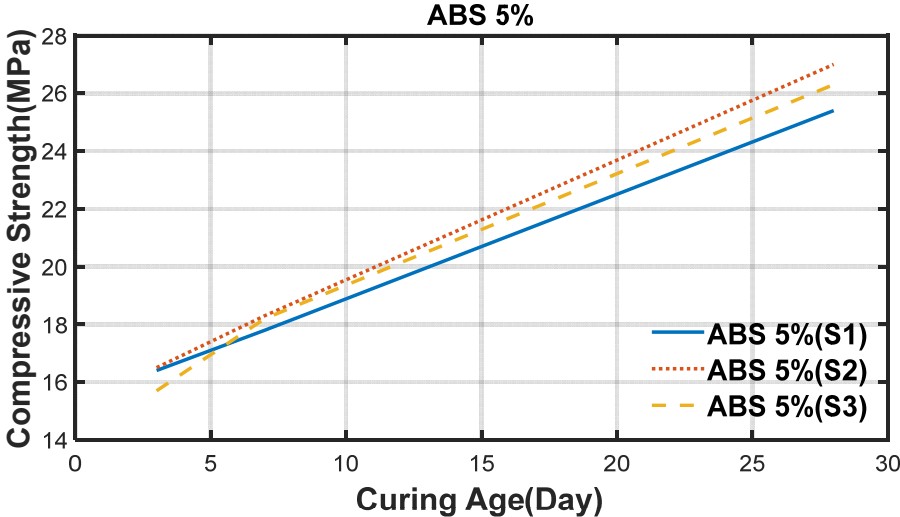

**Figure 8.** Graph of strength measurement results between ABS (5%) specimens.

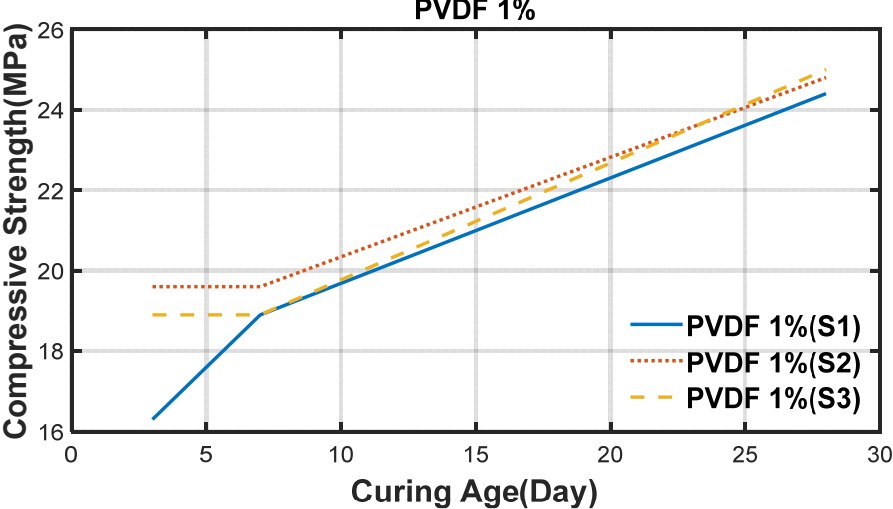

**Figure 9.** Graph of strength measurement results between PVDF (1%) specimens.

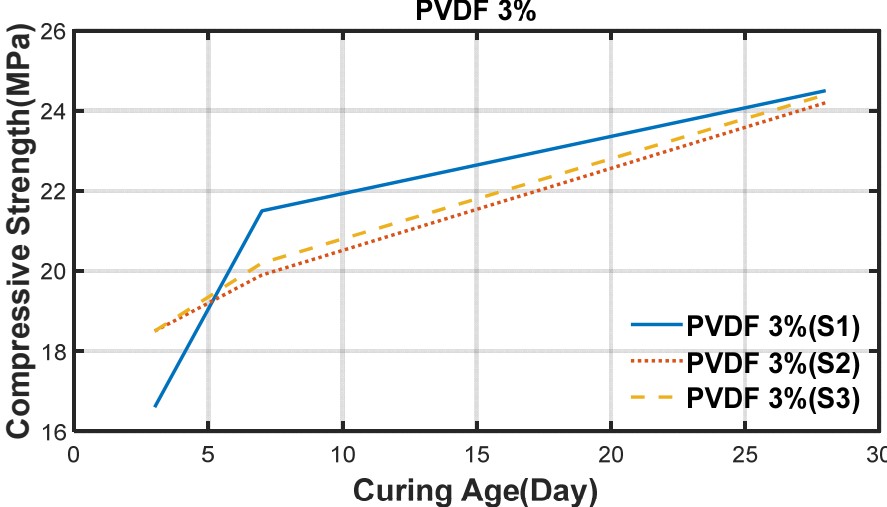

**Figure 10.** Graph of strength measurement results between PVDF (3%) specimens.

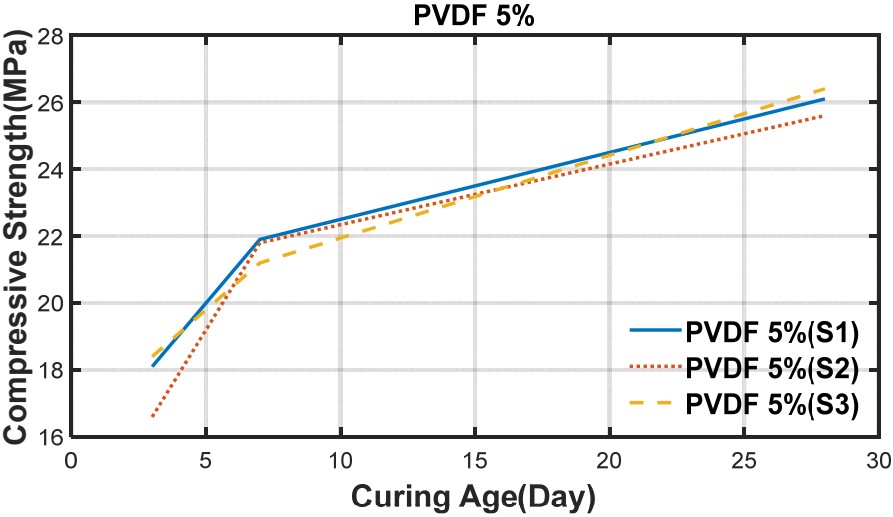

**Figure 11.** Graph of strength measurement results between PVDF (5%) specimens.

By using the compressive strength tests, it was confirmed that all the specimens had a generally uniform pattern and increased with age. In addition, it was confirmed that the tendency of the average strength value and the tendency of each specimen were consistent, and thus it was verified that comparison using the average strength value between the OPC, ABS, and PVDF specimens was possible.

### 3.3. Experimental Discussion

Table 7 and Figure 12 show the comparison of the average strength according to the addition of the mixture and its percentage.

**Table 7.** Compressive strength by mixture addition and percentage.

| Mixture | Percentage (%) | Age (Days) | | |
|---|---|---|---|---|
| | | 3 | 7 | 28 |
| OPC | - | 15.9 MPa | 21.2 MPa | 25.6 MPa |
| ABS | 1 | 16.9 MPa | 21.2 MPa | 25.6 MPa |
| | 3 | 17.2 MPa | 18.3 MPa | 25.8 MPa |
| | 5 | 16.2 MPa | 18.1 MPa | 26.2 MPa |
| PVDF | 1 | 19.1 MPa | 19.2 MPa | 24.7 MPa |
| | 3 | 17.9 MPa | 20.5 MPa | 24.4 MPa |
| | 5 | 17.7 MPa | 21.6 MPa | 26 MPa |

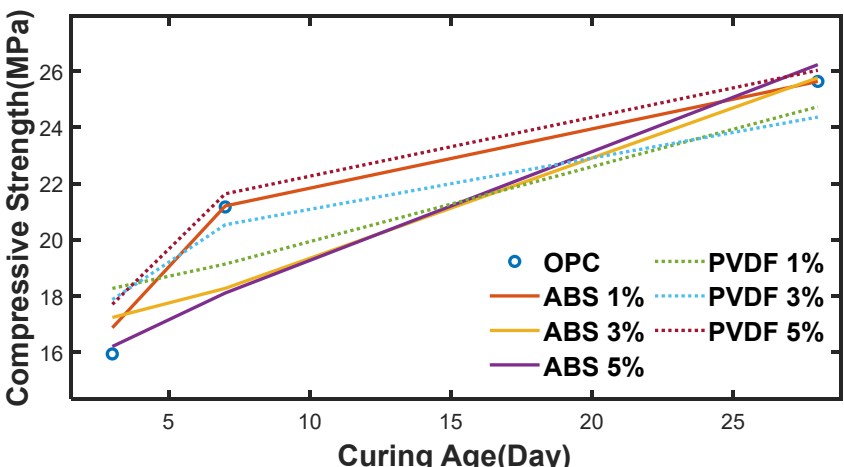

**Figure 12.** Comparison of compressive strength of specimens.

It can be seen that when ABS replaced 1% of cement, the strength of the specimen was almost the same as that of OPC concrete; however, when 3% and 5% ABS replaced the cement, it was confirmed that the strength in the first seven days was 3 MPa lower. This is suggested to be due to the replaced ABS powders interfering with the initial hydration reaction of cement. However, the strength after 28 days was high enough to use for concrete. Hence, it was confirmed that the strength of the concrete did not decrease even when ABS powder was added.

Moreover, it can be observed that the initial strengths of the PVDF specimens on the third day were all higher than that of the OPC specimen. It is assumed that the addition of PVDF powder absorbed a lot of water and resulted in high initial strength. In specimens to which 1% and 3% PVDF powder was added, it was confirmed that, except for the initial strength, the strength during the remaining days was generally low; indeed, the compressive strength on the 28th day was approximately 1 MPa lower than that of the OPC specimen. Conversely, it was confirmed that the specimen to which 5% PVDF powder was added had higher compressive strength than the OPC specimen at all ages. This phenomenon could be considered to be caused by the improved internal bonding strength due to the electric dipole of the replaced PVDF and the high moisture content of PVDF. According to the results of the experiments, the ABS and PVDF from the discarded membrane can be used as substitutes of cement to reduce carbon in regards to their compressive strength and workability.

## 4. Conclusions

This study considered reducing the amount of cement used in concrete by replacing it with discarded membrane resources. The characteristics of the age-specific strength were examined for a number of specimens with different mix ratios. A summary of this study is as follows.

1.  It was found that carbon emission reduction of approximately 50 kg $CO_2/m^3$ is possible through waste membrane recycling and cement replacement. This is equivalent to 10 pine trees absorbing 5.6 kg of $CO_2$ per year.
2.  To use the discarded membrane module as substitutes of cement, the outer casing (ABS) of membrane module and inner membrane (PVDF) should be made into powder form with a frozen pulverizing process.
3.  The concrete specimens were made with ABS and PVDF with 1 wt%, 3 wt%, 5 wt% to substitute cement, and the air contents test and slump test were conducted. According to the test results, the workability of ABS specimens was the same as the OPC specimen; for the PVDF specimen it was lower that OPC, but it was within the standard range.
4.  The compressive strength test was performed to find the applicability of ABS/PVDF concrete specimens. The compressive strengths of test specimens with ABS and PVDF were measured at the 3rd, 7th, and 28th days after pouring, and the strengths of OPC specimens were also measured for comparison with test specimens. As a result, the specimens to which 1% of ABS powder and 5% of PVDF powder was added had similar strength development to that of the OPC specimen; the numerical values were also confirmed to be almost identical.
5.  Furthermore, the specimens which contained 5% of PVDF showed higher strength than OPC specimens, caused by the improved internal bonding strength due to the electric dipole of the replaced PVDF and the high moisture content of PVDF.
6.  Therefore, if ABS powder is used to replace cement, it is judged that 1% replacement is possible, and in the case of PVDF powder, it is judged that it is possible to replace around 5%.

According to the results of the experiments, the ABS and PVDF from the discarded membrane can be substitutes of cement to reduce carbon in light of their compressive strength and workability.

**Author Contributions:** Conceptualization, J.K. ; methodology, J.K.; data curation, S.P.; writing—original draft preparation, S.P.; writing—review and editing, J.K.; visualization, S.P.; supervision, J.K.; project administration, J.K.; funding acquisition, J.K. All authors have read and agreed to the published version of the manuscript.

**Funding:** This research and APC funded by Ministry of Oceans and Fisheries (Development of Disaster Control and Aging Management Method for Port Infrastructure) (20210603).

**Informed Consent Statement:** Not applicable.

**Data Availability Statement:** Not applicable.

**Conflicts of Interest:** The authors declare no conflict of interest.

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
