# Peer review of "Performance Assessment of Concrete Using Discarded Membrane Filter Materials"

_water, doi:10.3390/w14142167_

Round 1

Reviewer 1 Report

Dear Editor:

Honestly, this manuscript was submitted previously to the "Material" Journal [Materials (ISSN 1996-1944)] "materials-1659592" by Four Authors "Sehwan Park, Minkyo Youm, Junkyeong Kim*, Yong‐Soo Lee". Firstly, the current authors, "Sehwan Park, Junkyeong Kim," must explain why they submit on their own? I reviewed it in two rounds as follows:

First round: Major (19 March 2022) (includes author's reply)

The manuscript studied the effect of replacing cement by 1%, 3%, and 5% waste ABS on the compressive strength of the normal strength concrete. This manuscript needs essential modifications as follows:

  • Line 80, "the prepared powder". The physical and chemical properties of the prepared powder must be presented in tables.
  • The mix proportions of mortar must be provided.
  • The compressive strength values in Table 1 must be reviewed.
  • Section 2.2.2 must be rephrased. Most of the tables and Eqs. 1 to 5 must be omitted.
  • Figure 3 is not necessary and should be omitted.
  • Lines 123 to 126 must be moved to Sec. 2.1.
  • Figure 5 duplicates Table 7, so it must be deleted.
  • The title of Section 3 must be renamed to be "Results and Discussion".
  • Section 3 must be extended, and the present results must be compared with the previous work.
  • The conclusion should be reduced.

Second round: Accept (7 April 2022)

The authors have successfully addressed all my comments.  Therefore, I recommend the publication of this manuscript.

However,

The second reviewer "Rejected it" in the second round for the following reasons:

"I reiterate my previous comments, the concept of the study seems very interesting, but the paper does not present sufficient quality to be published. The authors have made some changes suggested by the reviewers, improving the work, but I do not see a substantial improvement in terms of content that would justify its publication. I mean that they have not introduced new tests that evaluate more properties of these materials, in fact they have not even evaluated initial properties such as workability, viscosity, etc., when adding one type or another of powdered fibers; they have simply added a sentence in the conclusions that they will carry out aging tests in the future. But what about the possible effects on the texture or other physical properties of these materials, such as porosity, etc. Therefore, increasing the number of graphs to appear more than what was already presented in the previous version does not justify its approval.

I say the authors not to be discouraged by this decision because the idea is really good, but it needs to be completed with more essays to make the work more solid."

After reviewing the present version of the manuscript: My current decision is "Accept."

Reviewer 2 Report

In the Reviewer opinion the research paper entitled “Development and Performance Assessment of Concrete using Discarded Membrane Filter Materials” is poor.

This study proposes the development of an eco-friendly concrete by mixing waste membrane resources with concrete. The environmental pollution and wastage of resources due to incineration, and the enormous amount of carbon dioxide generated during cement production can be decreased by reducing the cement required when mixing concrete. The membrane module outer surface (acrylic butadiene styrene, ABS) and inner membrane (poly vinylidene fluoride, PVDF) were extracted from the waste membrane system and pulverized. The test specimens were then tested and compared with the reference concrete (ordinary Portland cement) specimen. The possibility of technological development of eco-friendly concrete using waste resources from membrane filtration facilities was verified.

Some comments which greatly enhance the understanding of the paper and its value are presented below. Specific issues that require further consideration are:

  1. The title of the manuscript is matched to its content, but it is too long.
  2. In the Reviewer’s opinion, the current state of knowledge relating to the manuscript topic has been presented, but the author's contribution and novelty are not enough emphasized.
  3. In the Reviewer’s opinion, the bibliography, comprising 14 references, is not representative.
  4. Please improve the quality of drawings.
  5. An analysis of the manuscript content and the References shows that the manuscript under review constitutes a summary of the Author(s) achievements in the field. However, the introduction needs more attention.
  6. Conclusion needs to be more revised and extended.
  7. Article has serious flaws, additional experiments needed, research not conducted correctly.
  8. In the Reviewer’s opinion the manuscript cannot be published in the journal.

Round 2

Reviewer 2 Report

Authors corrected article follow to my suggestion. In my opinion should be published in the Journal.